# SPREADSHEETCODER: FORMULA PREDICTION FROM SEMI-STRUCTURED CONTEXT

## ABSTRACT

Spreadsheet formula prediction has been an important program synthesis problem with many real-world applications. Previous works typically utilize input-output examples as the specification for spreadsheet formula synthesis, where each input-output pair simulates a separate row in the spreadsheet. However, such a formulation does not fully capture the rich context in real-world spreadsheets. First, spreadsheet data entries are organized as tables, thus rows and columns are not necessarily independent from each other. In addition, many spreadsheet tables include headers, which provide high-level descriptions of the cell data. However, previous synthesis approaches do not consider headers as part of the specification. In this work, we present the first approach for synthesizing spreadsheet formulas from tabular context, which includes both headers and semi-structured tabular data. In particular, we propose SPREADSHEETCODER, a BERT-based model architecture to represent the tabular context in both row-based and column-based formats. We train our model on a large dataset of spreadsheets, and demonstrate that SPREADSHEETCODER achieves top-1 prediction accuracy of $42.51\%$, which is a considerable improvement over baselines that do not employ rich tabular context.

## 1 INTRODUCTION

Spreadsheets are ubiquitous for data storage, with hundreds of millions of users. Support for helping users write formulas in spreadsheets is a powerful feature for data analysis. Although spreadsheet formula languages are relatively simpler than general-purpose programming languages for data manipulation, writing spreadsheet formulas could still be tedious and error-prone for end users (Gulwani, 2011; Hermans et al., 2012b; Cheung et al., 2016). Systems such as FlashFill (Gulwani, 2011; Gulwani et al., 2012) help end-users perform string transformation tasks in spreadsheets using a few input-output examples by automatically synthesizing a program in a domain-specific language (DSL). Recently, several learning approaches based on different neural architectures have been developed for learning such programs from examples, and have demonstrated promising results (Parisotto et al., 2017; Devlin et al., 2017; Vijayakumar et al., 2018).

All these previous works formalize the spreadsheet program prediction problem as a *programming by example* task, with the goal of synthesizing programs from a small number of input-output examples. We argue that this choice engenders three key limitations. First, this setup assumes that each data row is independent, and each formula is executed on data cells of the same row. However, real spreadsheets are less structured than this. Data in spreadsheets is typically organized as semi-structured tables, and cells in different rows could be correlated. As shown in Figure 1, in the same table, different data blocks could have different structures, without a common schema. Formulas can take cell values in other rows as function arguments. Second, because spreadsheets are semi-structured, they also contain rich metadata. In particular, many spreadsheet tables include headers that provide high-level descriptions of the data, which could provide important clues for formula prediction. However, table headers are not utilized in prior work. Finally, programming-by-example methods output programs in a DSL, which is typically designed to facilitate synthesis, and is much less flexible than the language in which users write formulas. For example, the FlashFill DSL only covers a subset of spreadsheet functions for string processing, and it does not support rectangular ranges, a common feature of spreadsheet formulas. In contrast, spreadsheet languages also support a wide variety of functions for numerical calculation, while the argument selection is more flexible and takes the spreadsheet

table structure into account. In total, these limitations can compromise the applicability of such prior efforts to more diverse real-world spreadsheets and to richer language functionality.

Instead, we propose synthesizing spreadsheet formulas *without* an explicit specification. To predict a formula in a given cell, the context of data and metadata is used as an *implicit* (partial) specification of the desired program. For example (Figure 1b), if predicting a formula at the end of a column of numbers labeled "Score", and a cell in the same row contains the text "Total", this context might specify the user's intent to compute a column sum. Our problem brings several new challenges compared to related work in programming by example (Gulwani, 2011; Bunel et al., 2018; Balog et al., 2017), semantic parsing (Popescu et al., 2003; Zhong et al., 2017; Yu et al., 2018) and source code completion (Raychev et al., 2014; Li et al., 2018; Svyatkovskiy et al., 2019). Spreadsheet tables contain rich two-dimensional relational structure and natural language metadata, but the rows do not follow a fixed schema as in a relational database. Meanwhile, our tabular context is more ambiguous as the program specification, and the spreadsheet language studied in this work is more flexible than in the program synthesis literature.

In this paper, we present SPREADSHEETCODER, a neural network architecture for spreadsheet formula prediction. SPREADSHEETCODER encodes the spreadsheet context in its table format, and generates the corresponding formula in the target cell. A BERT-based encoder (Devlin et al., 2019) computes an embedding vector for each input token, incorporating the contextual information from nearby rows and columns. The BERT encoder is initialized from the weights pre-trained on English text corpora, which is beneficial for encoding table headers. To handle cell references, we propose a two-stage decoding process inspired by sketch learning for program synthesis (Solar-Lezama, 2008; Murali et al., 2018; Dong & Lapata, 2018; Nye et al., 2019). Our decoder first generates a formula sketch, which does not include concrete cell references, and then predicts the corresponding cell ranges to generate the complete formula.

For evaluation (Section 4), we construct a large-scale benchmark of spreadsheets publicly shared within our organization. We show that SPREADSHEETCODER outperforms neural network approaches for programming by example (Devlin et al., 2017), and achieves $42.51\%$ top-1 full-formula accuracy, and $57.41\%$ top-1 formula-sketch accuracy, both of which are already high enough to be practically useful. Moreover, SPREADSHEETCODER can predict cell ranges and around a hundred different spreadsheet operators, which is much more flexible than DSLs used in prior works. With various ablation experiments, we demonstrate that both implicit specification from the context and text from the headers are crucial for obtaining good performance.

## 2 PROBLEM SETUP

In this section, we discuss the setup of our spreadsheet formula prediction problem. We first describe the input specification, then introduce the language and representation for spreadsheet formulas.

**Input specification.** We illustrate the input context in Figure 1. The input context consists of two parts: (a) context surrounding the target cell (e.g., all cell values in rows 2–7, and columns A–D, excluding cell D4 in Figure 1a), and (b) the header row (e.g., row 1).

In contrast to prior programming-by-example approaches (Gulwani, 2011; Parisotto et al., 2017; Devlin et al., 2017; Vijayakumar et al., 2018), our input specification features (a) tabular input, rather than independent rows as input-output examples, and (b) header information. Tabular input is important for many cases where formulas are executed on various input cells from different rows and columns (Figure 1), and headers hold clues about the purpose of a column as well as its intended type, e.g, the header cell "Score" in Figure 1b is likely to indicate that the column data should be numbers.

Note that we do not include the intended *output* of the target cell in our input specification, for three reasons. First, unlike programming-by-example problems, we do not have multiple independent input-output examples available from which to induce a formula, so providing *multiple* input-output examples is not an option. Second, even for our single input instance, the evaluated formula value may not be known by the spreadsheet user yet. Finally, we tried including the intended formula execution *result* in our specification, but it did not improve the prediction accuracy beyond what the contextual information alone allowed.

**The spreadsheet language.** Our model predicts formulas written in the Google Sheets language[1]. Compared to the domain-specific language defined in FlashFill, which focuses on string transfor-

---

[1]Google Sheets function list: `https://support.google.com/docs/table/25273?hl=en`.

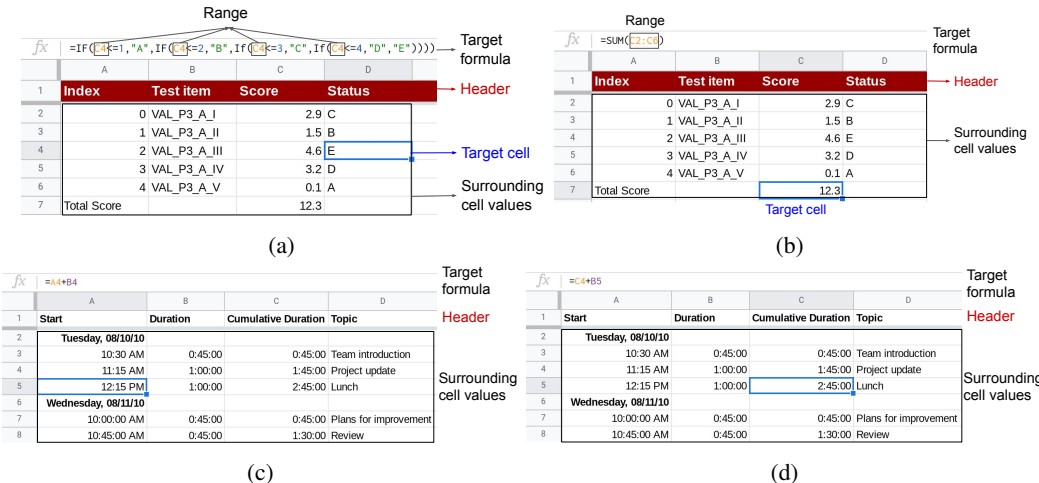

Figure 1: Illustrative synthetic examples of our spreadsheet formula prediction setup. (a): The formula manipulates cell values in the same row. (b): The formula is executed on the rows above. (c) and (d): Formulas involve cells in different rows and columns. The data value in the target cell is excluded from the input. All of these formulas can be correctly predicted by our model.

mations, the spreadsheet language supports a richer set of operators. Besides string manipulation operators such as CONCATENATE, LOWER, etc., the spreadsheet language also includes operators for numerical calculations (e.g., SUM and AVERAGE), table lookups (e.g., VLOOKUP) and conditional statements (IF, IFS). As will be discussed in Section 4, around a hundred different base formula functions appear in our dataset, many more than the operators defined in the FlashFill DSL. We limit our problem to formulas with references to *local* cells in a spreadsheet tab, thus we exclude formulas with references to other tabs or spreadsheets, and absolute cell ranges.

**Formula representation.** One of the key challenges in formula representation is how to represent cell references, especially ranges, which are prevalent in spreadsheet formulas. Naively using the absolute cell positions, e.g., A5, may not be meaningful across different spreadsheets. Meanwhile, a single spreadsheet can have millions of cells, thus the set of possible ranges is very large.

To address this, we design a representation for formula sketches inspired by prior work on sketch learning for program synthesis (Solar-Lezama, 2008; Murali et al., 2018; Dong & Lapata, 2018; Nye et al., 2019). A formula sketch includes every token in the prefix representation of the parse tree of the spreadsheet formula, except for cell references. References, which can be either a single cell or a range of cells, are replaced with a special placeholder RANGE token. For example, the sketch of the formula in Figure 1a is IF <= RANGE 1 "A" IF <= RANGE 2 "B" IF <= RANGE 3 "C" IF <= RANGE 4 "D" "E" $ENDSKETCH$, where $ENDSKETCH$ denotes the end of the sketch. Notice that the sketch includes literals, such as the constants 1 and "A".

To complete the formula representation, we design an intermediate representation for ranges, *relative* to the target cell. For example, B5 in Figure 1c is represented as $R$ R[0] C[1] $ENDR$ since it is on the next column but the same row as the target cell A5, and range C2:C6 in Figure 1b is represented as $R$ R[-5] C[0] $SEP$ R[-1] C[0] $ENDR$. The special tokens $R$ and $ENDR$ start and conclude a concrete range, respectively, and $SEP$ separates the beginning and end (relative) references of a rectangular multi-cell range.

A complete spreadsheet formula includes both the sketch and any concrete ranges; e.g., the formula in Figure 1b is represented as SUM RANGE $ENDSKETCH$ $R$ R[-5] C[0] $SEP$ R[-1] C[0] $ENDR$ EOF, where EOF denotes the end of the formula. In Section 3.2, we will discuss our two-stage decoding process, which sequentially predicts the formula sketch and ranges.

## 3 SPREADSHEETCODER MODEL ARCHITECTURE

In this section, we present our SPREADSHEETCODER model architecture for spreadsheet formula prediction. We provide an overview of our model design in Figure 2.

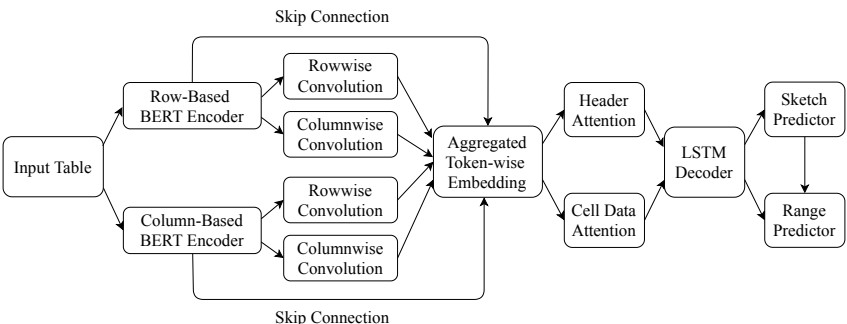

Figure 2: An overview of our model architecture.

### 3.1 TABULAR CONTEXT ENCODER

**Input representation.** Our model input includes the surrounding data values of the target cell as a table, and the first row is the header. When there is no header in the spreadsheet table, we set the header row to be an empty sequence. We include data values in cells that are at most $D$ rows and $D$ columns away from the target cell, so that the input dimension is $(2D+2) \times (2D+1)$, and we set $D = 10$ in our experiments.

**Row-based BERT encoder.** We first use a BERT encoder (Devlin et al., 2019) to compute a row-based contextual embedding for each token in the target cell's context. Since our $2D + 1 + 1$ rows contain many tokens and we use a standard BERT encoder of 512-token inputs, we *tile* our rows into bundles of three adjacent data rows, plus the header row, which is included in every bundle. Then we compute a token-wise BERT embedding for each bundle separately; the BERT weights are initialized from a pre-trained checkpoint for English. Specifically, in our experiments where $D = 10$, we concatenate all cell values for each row $i$ in the context into a token sequence $R_i$, which has length $L = 128$ (we trim and pad as needed). We combine rows in bundles $S_{rb} = [H_r, R_{3b-1}, R_{3b}, R_{3b+1}]$, for $b \in [-3, 3]$; here $H_r$ is the header row. We set the BERT segment IDs to 0 for the header tokens, and 1 for data tokens in each bundle. There are $2D + 1 = 21$ rows of context, so each of the 21 data rows is covered exactly once by the seven bundles. The header row is assigned a different BERT representation in each bundle. To obtain a single representation of the header row, we average per token across the embeddings from all of the bundles.

**Column-based BERT encoder.** As shown in Figure 1b, some formulas manipulate cells in the same column, in which case a column-based representation may be more desirable. Therefore, we also compute a column-based contextual embedding for all context tokens. We perform similar tiling as for the row-based BERT encoding, yielding column bundles $S_{cb}$ for $b \in [-3, 3]$. Unlike with row-wise tiling, where we include the header row $H_r$ with every bundle, for column-wise tiling we use the column of the target cell, $H_c = C_0$, as the "header column" in every bundle. After obtaining all token embeddings from this tiled computation by the BERT encoder, we discard token embeddings of $C_0$ in its role as header column, and only use its regular token embeddings from bundle $S_{c0}$.

**Row-wise and column-wise convolution layers.** Although the output vectors of BERT encoders already contain important contextual information, such as headers, nearby rows and columns, they still do not fully embed the entire input table as the context. To encode the context from more distant rows and columns, we add a row-wise convolution layer and a column-wise convolution layer on top of each BERT encoder. Specifically, the row-wise convolution layer has a kernel size of $1 \times L$, and the column-wise convolution layer has a kernel size of $(2D + 2) \times 1$ for row-based BERT, and $(2D + 1) \times 1$ for column-based BERT. In this way, the convolution layer aggregates across BERT embeddings from different bundles, allowing the model to take longer range dependencies into account. For each input token, let $e_b$ be its BERT output vector, $c_r$ be the output of the row-wise convolution layer, and $c_c$ be the output of the column-wise convolution layer. The final embedding of each input token is the concatenation of the BERT output and the output of convolution layers, i.e., $e = [c_r + c_c; e_b]$.

### 3.2 TWO-STAGE FORMULA DECODER

We train an LSTM (Hochreiter & Schmidhuber, 1997) decoder to generate the formula as a token sequence. Meanwhile, we use the standard attention mechanism (Bahdanau et al., 2015) to compute

two attention vectors, one over the input header, and one over the cell data. We concatenate these two attention vectors with the LSTM output, and feed them to a fully-connected layer with the output dimension $|V|$, where $|V|$ is the vocabulary size of formula tokens. Note that the token vocabularies are different for sketches (formula operators, literals, and special tokens) and ranges (relative row and column tokens and special range tokens). The output token prediction is computed with the softmax.

As mentioned in Section 2, we design a two-stage decoding process, where the decoder first generates the formula sketch, and then predicts the concrete ranges. In the first stage, the sketch is predicted as a sequence of tokens by the LSTM, and terminates when an $ENDSKETCH$ token is predicted. Then in the second stage, the range predictor sequentially generates formula ranges corresponding to each RANGE token in the sketch, and the prediction terminates when an EOF token is generated. Both sketch and range predictors share the same LSTM, but with different output layers.

## 4 EXPERIMENTS

We evaluate SPREADSHEETCODER on spreadsheet formula prediction tasks in different settings. We first describe our dataset, then introduce our experimental setup and discuss the results.

### 4.1 DATASET

We constructed our dataset from a corpus of Google Sheets documents publicly shared within our organization. Specifically, we collected 46K Google Sheets with formulas, and split them into 42K for training, 2.3K for validation, and 1.7K for testing. We filtered out formulas with cell references farther than 10 rows or columns from the target cell in either direction. Finally, we have 770K training samples, 42K for validation, and 34K for testing. In total, around 100 operators are covered in our output vocabulary. See Appendix C for details about dataset construction.

By default, each sample includes both the header row and surrounding data values of relative cell positions within $[-10, 10]$. Note that we do not include the data of the target cell, and we leave an empty value there. In Section 4.3, we also discuss settings when the model input does not include headers, and when we only include a few data rows above the target cell as the input context.

### 4.2 EVALUATION SETUP

**Metrics.** We evaluate the following metrics: (1) *Formula accuracy*: the percentage of predicted formulas that are the same as the ground truth. (2) *Sketch accuracy*: the percentage of predictions with the same formula sketches as the ground truth. As discussed in Section 2, formula sketches do not include ranges, but include both functions and literals. (3) *Range accuracy*: the percentage of predictions with the same ranges as the ground truth. Note that the order of predicted ranges should also be the same as the ground truth. In addition, the model may predict the ranges correctly even if the sketch prediction is wrong, and we provide an example in Figure 4b.

Note that our formula accuracy metric could be an underestimate of the semantic equivalence, because different spreadsheet formulas may be semantically equivalent. For example, to predict arguments for SUM and MULTIPLY, different orders of the cell ranges have the same meaning. However, it is hard to systematically define the semantic equivalence in our evaluation, because we aim to support a wide range of operators in the spreadsheet language. Some existing works on program synthesis have evaluated the semantic equivalence based on the execution results (Devlin et al., 2017; Bunel et al., 2018; Sun et al., 2018). However, it is hard to sample different input spreadsheets requiring the same formula, thus evaluating the execution accuracy is challenging. Therefore, we still focus on our current metric to measure the formula accuracy, where we compare whether the predicted formula is exactly the same as the single ground truth formula included in the spreadsheet.

**Model details.** For models with the BERT encoder (Devlin et al., 2019), including our full SPREAD-SHEETCODER model, we use the BERT-Medium architecture, and initialize from the English pre-trained model by default.[2] We compared our full model with several variants:

*(1) Different encoder architectures*: i) using a single BERT encoder, either row-based or column-based; ii) removing convolution layers, where the BERT output is directly fed into the decoder.

---

[2] We downloaded the pre-trained BERT from: https://github.com/google-research/bert.

*(2) Different decoding approaches*: We compare our proposed two-stage decoding discussed in Section 3.2 to a simpler model that uses the same predictor for both the sketch and ranges, with a single joint output vocabulary for both.

*(3) Different model initialization*: When not using the pre-trained BERT model weights, we randomly initialize BERT encoders. This tests whether pre-training on generic natural language text is useful for our spreadsheet data.

We compare to previous approaches for related program synthesis tasks. First, we evaluate RobustFill, which demonstrates the state-of-the-art performance on string manipulation tasks for Excel spreadsheets (Devlin et al., 2017). Specifically, RobustFill is designed for FlashFill, where each formula is executed on a single data row, thus each row is independently fed into a shared encoder. We defer the details of RobustFill model architecture to Appendix A. We trained two variants of RobustFill on our dataset: one encodes each row independently, and another encodes each column independently, denoted as *row-based RobustFill* and *column-based RobustFill* respectively. In addition, we compared to a baseline that does not utilize any input context, thus the model only includes the LSTM decoder, similar to prior work on language modeling (Sundermeyer et al., 2012; Karpathy et al., 2015).

## 4.3 RESULTS

In this section, we present the results using different variants of spreadsheet contexts as the model inputs. For all settings, we perform a beam search during the inference time, with a beam size 64.

### 4.3.1 RESULTS WITH THE FULL INPUT CONTEXT

Using both headers and the full surrounding data cell values as the model input, we present the formula accuracy in Table 1, where top-$k$ accuracy measures how often the ground truth appears in the top $k$ predictions using beam search. Compared to the model without the input context, all other models are able to use the contextual data to provide more accurate predictions. In particular, our full model achieves over $40\%$ top-1 full formula prediction accuracy, which is 4 times as high as the model without context. We also observe that the full SpreadsheetCoder model has much better accuracy than either of the RobustFill models, demonstrating that our model is more capable of leveraging the implicit specification provided by the tabular context.

**Different encoder architectures.** Appropriately encoding the input context is important. Comparing with RobustFill models, we observe that it is beneficial to model the dependency among different rows and columns, instead of encoding each row or column independently. Meanwhile, adding convolution layers brings additional performance gain, because it enables the representations of each input token to aggregate broader contextual information beyond a few nearby rows or columns, i.e., 3 for our BERT encoders as discussed in Section 3.1. Finally, although models representing the input context as column-based tables generally perform worse than those using row-based tables, including both row-based and column-based encoders improves the overall accuracies by 2–3 percentage points. Note that the improvement is not due to the larger model size: to test this, we trained row-based and column-based BERT models with the larger BERT-base and BERT-large architectures, but the results were no better, while taking longer to train. In addition, initializing from pre-trained BERT encoders increases the formula accuracy by around 10 percentage points, suggesting that although spreadsheet headers are generally short natural language phrases, pre-training on a large-scale text corpus with much more complex text still enables the model to better understand the spreadsheet context.

**Breakdown analysis of sketch and range prediction.** We present the sketch and range accuracies in Table 2. On the one hand, sketch accuracies are generally much higher than range accuracies, since formulas are more likely to share common sketches with similar spreadsheet context, while range prediction requires a more careful investigation of the table structure. On the other hand, sketch prediction becomes more challenging when literals are included. In Figure 4a, we present a sample prediction with the correct sketch but the wrong range. Specifically, the model could easily infer that the formula should call a SUM function, since it is a common prediction given the input token "Total". However, the model wrongly selects all cells above as the function argument, and ignores the fact that the cell B5 is already the sum of cells B2−B4, indicated by the text "Total Cost" in cell A5. Figure 4b shows a prediction with the correct range but the wrong sketch, where the predicted formula misses a "/" as an argument to the string concatenation operator "&". Two-stage decoding disentangles the generation of sketches and ranges, so that the two predictors could focus on addressing different difficulties in formula prediction, and this mechanism provides additional accuracy improvement.

Table 1: Formula accuracy on the test set. "−" means the corresponding component is removed from our full model.

| | Approach | Top-1 | Top-5 | Top-10 |
|---|---|---|---|---|
| | Full Model | **42.51%** | **54.41%** | **58.57%** |
| − | Column-based BERT | 39.42% | 51.68% | 56.50% |
| − | Row-based BERT | 20.37% | 40.87% | 48.37% |
| − | Convolution layers | 38.43% | 51.31% | 55.87% |
| − | Two-stage decoding | 41.12% | 53.57% | 57.95% |
| − | Pretraining | 31.51% | 42.64% | 49.77% |
| | Row-based RobustFill | 31.14% | 40.09% | 47.10% |
| | Column-based RobustFill | 20.65% | 39.69% | 46.96% |
| | No context | 10.56% | 23.27% | 31.96% |

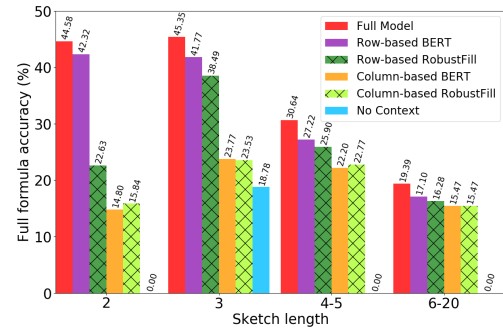

Figure 3: Top-1 formula accuracies for different formula sketch lengths.

Table 2: Sketch and range accuracy on the test set.

(a) Sketch accuracy.

| | Approach | Top-1 | Top-5 | Top-10 |
|---|---|---|---|---|
| | Full Model | **57.41%** | **72.04%** | **78.52%** |
| − | Column-based BERT | 55.50% | 70.88% | 77.73% |
| − | Row-based BERT | 27.49% | 61.95% | 73.95% |
| − | Convolution layers | 53.68% | 69.38% | 75.67% |
| − | Two-stage decoding | 56.47% | 72.02% | 78.30% |
| − | Pretraining | 41.26% | 64.67% | 76.36% |
| | Row-based RobustFill | 40.23% | 61.50% | 72.20% |
| | Column-based RobustFill | 29.50% | 59.97% | 71.31% |
| | No context | 25.19% | 47.08% | 52.70% |

(b) Range accuracy.

| | Approach | Top-1 | Top-5 | Top-10 |
|---|---|---|---|---|
| | Full Model | **46.93%** | **59.60%** | **63.51%** |
| − | Column-based BERT | 43.60% | 57.12% | 62.27% |
| − | Row-based BERT | 22.57% | 47.84% | 55.29% |
| − | Convolution layers | 42.84% | 56.64% | 61.03% |
| − | Two-stage decoding | 44.59% | 58.52% | 62.48% |
| − | Pretraining | 36.03% | 49.85% | 54.71% |
| | Row-based RobustFill | 33.88% | 48.16% | 54.83% |
| | Column-based RobustFill | 23.97% | 47.09% | 52.75% |
| | No context | 11.80% | 25.54% | 38.07% |

**Prediction on formulas with different sketch lengths.** We present the top-1 formula accuracy on formulas with different sketch lengths in Figure 3. Note that we exclude the $ENDSKETCH$ token from length calculation. First, all models achieve higher performance on formulas with sketch lengths of 2–3 than longer formulas. It is harder to make exactly the same prediction as the ground truth when the formula becomes longer, especially given that the input context is often an ambiguous specification for formula prediction. Fortunately, users typically do not need to write complicated formulas for spreadsheet data manipulation. Specifically, 85% of our collected formulas have sketch lengths of 2–3. Despite the performance degradation, our full model consistently performs better than other models on formulas with different sketch lengths.

### 4.3.2 THE EFFECT OF HEADER INFORMATION

In this section, we evaluate the effect of including the header row as the model input, which usually provides a short description of the table in natural language. For all models, we remove the headers from the context by replacing the header tokens with empty values. Thus the models can only use surrounding data cells as the spreadsheet context.

In Table 3, we observe a notable accuracy drop compared to Table 1, indicating that leveraging headers is critical. Figure 6a shows an example that can be correctly predicted by our full model, but is wrongly predicted by the model without input headers. We can observe that without the header "Average", it is much harder to figure out that the formula should call the AVERAGE function instead of a division. Interestingly, without input headers, using row-based or column-based table representation no longer makes much difference. However, our tabular input context encoders still perform better than RobustFill models, suggesting the importance of modeling the dependency among different rows and columns. In addition, initializing from pre-trained BERT model weights does not improve the results, and even slightly hurts the performance. The main reason is that the cell data values are mostly numeric and string literals. Breakdown results are deferred to Appendix B.

### 4.3.3 RESULTS IN THE FLASHFILL-LIKE SETTING

Finally, we conduct experiments in the FlashFill-like setting, where formulas are always executed on cells in the same row. In total, 2.5K formulas in the test set only include cells with the relative row position R[0], which constitute around 73% of the test set. More details are in Appendix D.

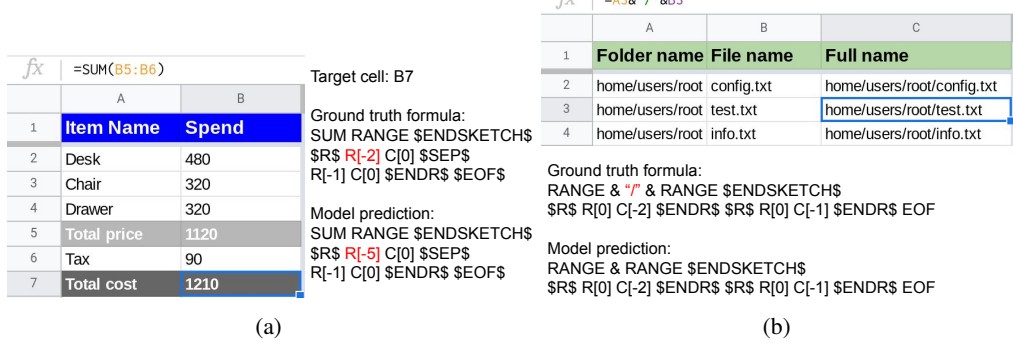

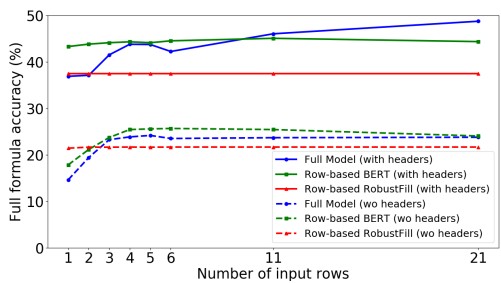

(a)                                                    (b)

Figure 4: Examples of wrong formula predictions by our full model. (a) The sketch prediction is correct, but the range is wrong. (b) The range prediction is correct, but the sketch is wrong. These are synthetic examples for illustrative purposes.

Table 3: Formula accuracy on the test set, excluding headers in the context. Corresponding results with headers are in Table 1.

| Approach | Top-1 | Top-5 | Top-10 |
|---|---|---|---|
| Full Model | 20.47% | 40.23% | 47.40% |
| − Column-based BERT | 20.63% | 40.40% | **48.70%** |
| − Row-based BERT | 20.38% | 40.11% | 47.88% |
| − Pretraining | **20.94%** | **40.64%** | 48.51% |
| Row-based RobustFill | 19.02% | 33.60% | 37.38% |
| Column-based RobustFill | 17.64% | 30.45% | 36.79% |
| No context | 10.56% | 23.27% | 31.96% |

Figure 5: Top-1 formula accuracy in the FlashFill-like setting, with different number of input rows.

In Figure 5, we present the top-1 formula accuracies with different numbers of input data rows. We observe that even for spreadsheet formulas that only refer to cells in the same row, our models with tabular input encoders still perform better. In particular, with the increase of the number of input data rows, the accuracy of the RobustFill model does not show much improvement, while the accuracies of the other two models increase considerably, especially our full model. This demonstrates that our model could better utilize the available cell data context for prediction. Figure 6b shows a formula that can be correctly predicted by our model when the full input context is given, but is wrongly predicted when the input only contains the header row and one data row. This example shows that understanding the cell data is especially important when the header is not informative enough. Notice that including only a few input rows or columns does not fit our encoder design well, since our BERT encoders simultaneously embed 3 data rows at a time, while the RobustFill model independently encodes each row by design. This could be the main reason why models with BERT-based encoders may perform worse than RobustFill when less than 3 data rows are presented. In addition, including headers still consistently provides a significant performance gain.

## 5 RELATED WORK

In this section, we present a high-level overview of the related work, and we defer a more in-depth discussion to Appendix A. *Program synthesis* has been a long-standing challenge, and various types of specifications have been discussed, including input-output examples (Gulwani et al., 2012; Balog et al., 2017; Bunel et al., 2018; Bavishi et al., 2019; Shin et al., 2018; Chen et al., 2019), natural language descriptions (Gulwani & Marron, 2014; Yu et al., 2018; Yin et al., 2018; Lin et al., 2018), and images (Wu et al., 2017; Liu & Wu, 2019; Sun et al., 2018). In particular, the FlashFill benchmark (Gulwani et al., 2012) is the most related to our task, and their goal is to generate string transformation programs to manipulate the Excel spreadsheet data, given input-output examples as the specification. Various neural network approaches have been proposed for FlashFill (Parisotto et al., 2017; Devlin et al., 2017; Vijayakumar et al., 2018). On the other hand, Nlyze (Gulwani & Marron, 2014) translates natural language specifications to programs in an SQL-like DSL for spreadsheet data

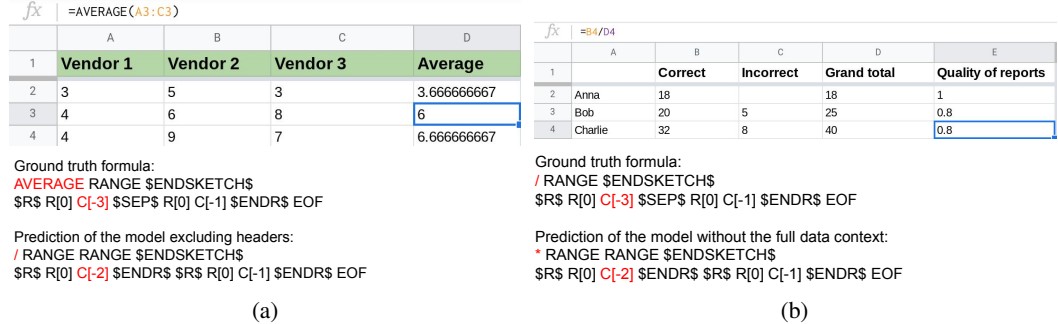

Figure 6: Examples of formulas that are correctly predicted by our full model with the full context, but wrongly predicted with missing context. (a) The wrong prediction when the model input does not include headers. Note that the model with headers predicts it correctly even if only one data row is provided. (b) The wrong prediction when the model input only includes headers and one data row. These are synthetic examples for illustrative purposes.

manipulation; and Autopandas (Bavishi et al., 2019) synthesizes dataframe transformation functions implemented with the Python Pandas library, given input-output dataframe examples. The spreadsheet formula prediction task in our work considers the semi-structured tabular spreadsheet context as the specification, rather than standardized input-output examples or natural language descriptions. Therefore, our formula specifications are more ambiguous and diverse. Furthermore, we show that including the header information is a key factor to improving the formula prediction performance.

In terms of the model input format, our spreadsheet formula prediction task is related to existing benchmarks on *semantic parsing* over a tabular database (Iyyer et al., 2017; Zhong et al., 2017; Yu et al., 2018). Various approaches have been proposed for these tasks (Liang et al., 2018; Wang et al., 2020; Yin et al., 2020; Herzig et al., 2020). There are two key differences between these works and ours. First, their program specification contains a natural language question, while our work predicts spreadsheet formulas based on the tabular context only. Therefore, our input specification is much more ambiguous. Meanwhile, our spreadsheet tables are typically less structured than the database tables. As shown in Figure 1, spreadsheet tables do not necessarily satisfy a consistent row-based schema, and data cell values may be dependent on cells from other rows.

Our spreadsheet formula prediction problem is also related to *code completion* tasks (Raychev et al., 2014; Li et al., 2018; Svyatkovskiy et al., 2019; 2020; Svyatkovskoy et al., 2020). Specifically, the goal of code completion tasks is to synthesize the subsequent program tokens given the code context, while we aim to generate the formula in the cell with the missing value to complete the spreadsheet. However, instead of providing a token sequence to represent the code context, our data context is a semi-structured table, where data values in different cells are connected in a two-dimensional space.

## 6 CONCLUSION

We presented the first technique to synthesize spreadsheet formulas given a tabular context, including both headers and cell values. In particular, we develop SPREADSHEETCODER, a BERT-based model to capture the two-dimensional relational structure of the spreadsheet context, which are typically semi-structured tables. We demonstrate that incorporating the table headers significantly facilitates the prediction. Furthermore, modeling the dependency among cells of different rows and columns is important for generating formulas in real-world spreadsheets with diverse table structures.

There are a number of promising directions for future research about spreadsheet applications. First, developing a paradigm for pre-training on spreadsheet data could enable the encoder to be more specialized for spreadsheet applications. Second, we could infer more fine-grained knowledge of the table structure from the spreadsheet format information, such as colors and fonts, which could be utilized to develop more advanced encoder architectures. Finally, we could also extend our approach to support more spreadsheet applications, such as bug detection and clone detection.

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

## A    AN EXTENDED DISCUSSION OF RELATED WORK

Various neural network approaches have been proposed for the FlashFill benchmark (Parisotto et al., 2017; Devlin et al., 2017; Vijayakumar et al., 2018). Specifically, both R3NN (Parisotto et al., 2017) and RobustFill (Devlin et al., 2017) are purely statistical models, and RobustFill performs better. In a RobustFill model, each formula is executed on a single data row, thus each row is independently fed into a shared encoder. Afterwards, at each decoding step, a shared LSTM decoder generates a hidden

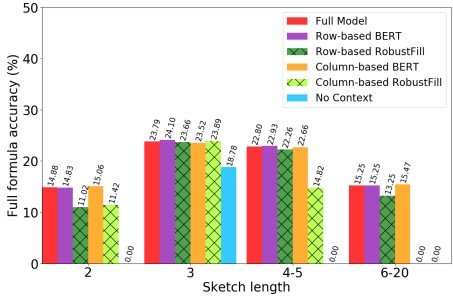

Figure 7: Top-1 formula accuracies for different sketch lengths, excluding headers in the context.

Table 4: Breakdown accuracies on the test set, excluding headers in the context.

(a) Sketch accuracy.

| Approach | Top-1 | Top-5 | Top-10 |
|---|---|---|---|
| Full Model | 28.33% | **62.55%** | 72.89% |
| − Column-based BERT | 28.40% | 61.60% | **74.92%** |
| − Row-based BERT | 27.71% | 60.84% | 73.43% |
| − Pretraining | **28.78%** | 62.37% | 74.61% |
| Row-based RobustFill | 25.78% | 42.66% | 50.17% |
| Column-based RobustFill | 26.15% | 47.78% | 57.72% |
| No context | 25.19% | 47.08% | 52.70% |

(b) Range accuracy.

| Approach | Top-1 | Top-5 | Top-10 |
|---|---|---|---|
| Full Model | 22.60% | 47.11% | 53.84% |
| − Column-based BERT | 22.82% | **47.76%** | **54.98%** |
| − Row-based BERT | 22.47% | 46.14% | 54.51% |
| − Pretraining | **23.48%** | 47.27% | 54.59% |
| Row-based RobustFill | 21.01% | 38.21% | 43.89% |
| Column-based RobustFill | 21.27% | 37.80% | 43.77% |
| No context | 11.80% | 25.54% | 38.07% |

state per data row, which are then fed into a max pooling layer. Finally, the pooled hidden state is fed into a fully-connected layer to predict the formula token. On the other hand, in (Vijayakumar et al., 2018), they design a neural network to guide the deductive search performed by PROSE (Polozov & Gulwani, 2015), a commercial framework for input-output program synthesis. A recent work proposes neural-guided bottom-up search for program synthesis from input-output examples, and they extend the domain-specific language of FlashFill to support more spreadsheet programs (Odena et al., 2020).

Besides formula prediction, some previous work has studied other applications related to spreadsheets, including smell detection (Hermans et al., 2012a; Cheung et al., 2016; Singh et al., 2017; Azam et al., 2019), clone detection (Hermans et al., 2013; Dou et al., 2016; Zhang et al., 2020), and structure extraction for spreadsheet tables (Dong et al., 2019a;b). Our proposed encoder architecture could potentially be adapted for these spreadsheet tasks as well, and we leave it for future work.

## B    MORE EXPERIMENTAL RESULTS

For the setting where the model input does not include headers, corresponding to Table 3 in Section 4.3.2, we present the sketch and range accuracies in Table 4, and the breakdown accuracies on formulas of different sketch lengths in Figure 7. We observe that the performance degradation is more severe for formulas of sketch lengths 2–3.

## C    DATASET CONSTRUCTION

Although in principle, our model could generate formulas using any operator in the spreadsheet language, some kinds of value references are impossible to predict from local context, thus we remove formulas with such values from our dataset. Specifically, we exclude formulas that use the HYPERLINK function with a literal URL, since those are merely "stylistic" formulas that perform no computation beyond presenting a URL as a clickable link. As discussed in Section 2, we also filtered out formulas with cross-references from other tabs or spreadsheets. In total, the formulas filtered out after these two steps constitute around 40% of all formulas. We further filtered out formulas with cell references farther than 10 rows or columns from the target cell in either direction, and formulas with absolute cell ranges. In this way, about 45% of the original set of formulas are kept in our dataset.

Meanwhile, we observe that some spreadsheets may have tens of thousands of rows including the same formula, and including all of them in the dataset could bias our data distribution. Therefore, when multiple rows in the same spreadsheet table include the same formula in the same column, we keep the first 10 occurrences of such a formula, and create one data sample per formula. In this way, we extract around 800K formulas from 20M formulas before this filtering step.

About the length distribution of target spreadsheet formulas, about 32% formulas have sketch lengths of 2, 53% formulas have sketch lengths of 3, 11% formulas have sketch lengths of 4-5, and 4% formulas have sketch lengths of at least 6. As discussed in Section 2, even if the formula sketches are mostly short, it is still challenging to generate the full formulas correctly. For example, the formula in Figure 1b is represented as SUM RANGE $ENDSKETCH$ $R$ R[-5] C[0] $SEP$ R[-1] C[0] $ENDR$ EOF, which has a sketch length of 2, but the full formula length is 10 if excluding the EOF token for length calculation.

Among all spreadsheet formulas included in our filtered dataset, we list the 30 most commonly used spreadsheet functions and operators with their types [3] as follows: SUM (Math), + (Operator, equivalent to ADD), − (Operator, equivalent to MINUS), * (Operator, equivalent to MULTIPLY), / (Operator, equivalent to DIV), & (Operator, equivalent to CONCAT), AVERAGE (Statistical), LEN (Text), UPLUS (Operator), STDEV (Statistical), COUNTA (Statistical), MAX (Statistical), LEFT (Text), IFERROR (Logical), ABS (Math), MEDIAN (Statistical), UMINUS (Operator), CONCATENATE (Text), ROUND (Math), WEEKNUM (Date), AVERAGEA (Statistical), MIN (Statistical), COUNT (Statistical), TRIM (Text), COS (Math), SIN (Math), SINH (Math), TODAY (Date), IF (Logical), MONTH (Date). We observe that most of these functions and operators are for mathematical calculation, statistical computation, and text manipulation. However, people also write conditional statements, and spreadsheet formulas for calculating the dates.

## D    More Discussion of the FlashFill-like Setting

Following prior work on FlashFill (Devlin et al., 2017; Parisotto et al., 2017; Vijayakumar et al., 2018), we evaluate model performance when different numbers of data rows are presented to the model as input. Specifically, when the input includes 1–11 data rows, we grow the input from the target row upward. Our full data context includes 21 data rows, with 10 rows above the target cell, 10 rows below the target cell, and 1 row where the target cell locates. Consistent with prior work, when we vary the number of input data rows during inference, we always evaluate the same model trained with the full data context including 21 data rows. Since RobustFill independently encodes each row, it supports variable number of input rows by design. For our models with the tabular input representation, we set the rows to be empty when they are out of the input scope, and apply a mask to indicate that the corresponding data values are invalid.

## E    Implementation Details

**Data preprocessing.**    The content in each cell includes its data type and value, and we concatenate them as a token sequence. For example, A2 in Figure 1a is represented as num 0. As discussed in Section 3.1, we concatenate all cell values in the same row as a token sequence, where values of different cells are separated by the [SEP] token. Each data row fed into the model includes $L = 128$ tokens, and when the concatenated token sequence exceeds the length limit, we discard cells that are further away from the target cell. For column-wise representation, we produce token embeddings independently for each column-wise bundle $S_{cb} = [H_c, C_{3b-1}, C_{3b}, C_{3b+1}]$ for $b \in [-3, 3]$, where $C_i$ is a token sequence produced by concatenating all tokens of the cells in column $C_i$. We perform the header detection according to the spreadsheet table format, i.e., we recognize the first row of a table as the header when it is frozen. Though some spreadsheet tables may include header-like descriptions in the leftmost column, e.g., "Total Score" in Figure 1a, we only extract headers as a row, to ensure the precision of header detection.

**Output vocabulary construction.**    To construct the output formula token vocabulary, we filtered out tokens that appear less than 10 times in the training set, so that the vocabulary contains 462

---

[3]The function types are based on the Google Sheets function list here: https://support.google.com/docs/table/25273?hl=en.

tokens, out of 2625 tokens before filtering. In total, around a hundred operators are covered in our output vocabulary, including 82 spreadsheet-specific functions, and other general-purpose numerical operators (e.g., +, −).

**Hyper-parameters.** The formula decoder is a 1-layer LSTM with the hidden size of 512. We train the model with the Adam optimizer, with an initial learning rate of 5e-5. We train models for 200K minibatch updates, with a batch size 64. We set the dropout rate to be 0.1 for training. The norm for gradient clipping is 1.0.

