# OpenReview forum: "SpreadsheetCoder: Formula Prediction from Semi-structured Context"
_ICLR.cc/2021/Conference — Reject_

### Official Review · AnonReviewer1 · 2020-10-27

**Rating:** 7
**Confidence:** 5

**Review:**

### Paper summary
This paper addresses the problem of inferring spreadsheet formula from surrounding cell values and headers (i.e. brief description of each column). Compared to the standard programming-by-example line of work, which concerns synthesizing a program from a set of independent input/output examples, the problem setup proposed in this paper is more realistic and allows a model to synthesize a program by incorporating the contextual information. To this end, the paper proposes a framework that is specially designed to encode the header and the data from multiple rows and columns. To alleviate the difficulty of decoding a program from scratch, it proposes to first produce a program sketch that consists of formula operators and literals, and then predict the relative range which specifies where the formula operators should be applied. The experiments compare the performance of baselines as well as the proposed framework and its variations. The experimental results show that the proposed framework outperforms the rest and justify many design choices (e.g. leveraging a pre-trained BERT model, predicting a program sketch first, encoding rows and columns independently, etc.) I believe this work proposes an interesting and promising research problem as well as a framework that can achieve reasonable performance. Therefore, I vote for acceptance.

### Paper strengths
**motivation & problem setup**
The motivation for incorporating contextual information for programming by example is convincing. I believe this will make the neural program synthesis line of work more applicable to real-world problems.

**novelty & technical contribution**
- As far as I am concerned, the problem setup is novel, since it includes header information, does not consider different rows independently, predicts a formula/program in a specified target cell, and allows to reference multiple cells.
- Leveraging a pre-trained BERT model seems crucial (~10% performance gap).
- Encoding row and column information separately and merging them with convolutions is compelling.
- Predicting a program sketch first to alleviate the difficulty of sequentially producing each token of a program is convincing. This paper presents an effective way to implement this idea.

**clarity**
The writing is very clear and the figures illustrate the ideas well. Also, the organization of the paper is easy to follow.

**ablation study**
Ablation studies are comprehensive. The proposed framework consists of multiple components. The provided ablation studies help analyze the effectiveness of each of them, including
- using a row BERT encoder only / using a column BERT encoder only / removing convolution layers that merge BERT's output
- decoding a program sketch and its range altogether (without the proposed two-stage decoding)
- using a BERT model that is not pre-trained on text corpora

**experimental result**
The presentation of the experimental results is very clear. The authors present insightful information, which includes:
- top-n accuracy
- performance vs. different program sketch lengths
- sketch only accuracy and range only accuracy
The analysis of the experiment results is detailed and insightful.

### Paper weaknesses
**formula accuracy**
If I understand correctly, it is possible to synthesize a formula differently from how the ground truth formula is written. If this is the case, I wonder how formula accuracy is computed as one would need to enumerate all the possible ways of writing a particular formula to do so.

**pre-trained BERT for headers only**
Given the sampled shown in the paper, I am not sure if it is good to employ a pre-trained BERT to encode the content of the spreadsheet except for the header. Therefore, I would like to know if the authors have tried to use a pre-trained BERT for encoding headers only while using a BERT learning from scratch to encode the rest of the spreadsheet.

**noisy data/header**
I wonder how the proposed framework can deal with headers that do not properly/correctly indicate the meaning of the data.

**dataset availability & more samples**
It would be great if the dataset was provided so that the readers can better judge the difficulty of the problem and the performance of the models. Even if it is not possible to make the dataset publicly available, it would be better if a set of randomly selected data points (spreadsheet + target formula) were included. Also, providing some statistics of the dataset would be helpful, such as the length distribution of the target formula, the most commonly used operators, etc.

**reproducibility**
Given the clear description in the main paper and the details provided in the appendix, I believe implementing the proposed framework is possible. Yet, without access to the dataset, it is still impossible to reproduce the results.

**related work**
While the related work sections in the main paper and the appendix sufficiently cover most of the relevant works, I believe this paper can still be benefit from including the following papers
- Improving Neural Program Synthesis with Inferred Execution Traces
- Execution-Guided Neural Program Synthesis
- Neural Scene De-rendering
- Learning to Describe Scenes with Programs
- Neural Program Synthesis from Diverse Demonstration Videos
- NL2Bash: A Corpus and Semantic Parser for Natural Language Interface to the Linux Operating System

---

> ### Author Response · Authors · 2020-11-21
> **Clarification and discussion**
>
> Thanks for appreciating our work and your insightful comments! We have incorporated your comments in our revision, and we respond to your questions and comments below.
>
> 1. Formula accuracy
>
> We agree that for the same input specification, it is possible to synthesize different semantically equivalent formulas. To evaluate the formula accuracy, we compare whether the predicted formula is exactly the same as the single ground truth formula that is included in the spreadsheet. Therefore, we consider any other predicted formula that is different from the ground truth as a wrong, even if it is semantically equivalent to the ground truth. Note that it is hard to systematically define the semantic equivalence in our evaluation, because we aim to support a wide range of operators in the spreadsheet language. Therefore, we still focus on our current metric to measure the formula accuracy, even if it is an under-estimate of the semantic equivalence. A similar discussion could be found in our response to R2 about the evaluation metric, and we have revised Section 4.2 in the paper to make this point clearer.
>
> 2. Pre-trained BERT for headers only
>
> Thanks for your great suggestion! We indeed have tried using a pre-trained BERT for headers and a randomly initialized BERT for data cells. However, the performance is 2~3% lower than using a pre-trained BERT for both. One possible reason could be that the model becomes harder to train because more BERT architectures are included in the model.
>
> 3. Noisy data/header
>
> For the dataset we collected, the headers of many spreadsheet tables only provide vague descriptions. For example, in Figure 1 (a), it is a non-trivial job to predict the formula based on the header “Status”, while our model still provides the correct prediction. In Figure 6 (b), we provide an example of wrong formula prediction when the model does not include the full tabular cell data as the formula specification, and the main reason could be that the header “Quality of reports” is not informative. The ambiguity of the formula specification, e.g., the noisy headers, is one key challenge for spreadsheet formula prediction. Our results show that by training on a large-scale dataset, our model appropriately captures the desired formula functionality from the headers, even when sometimes they do not explicitly convey the meaning of the data. We have moved the sample model predictions in the appendix to Section 4.3 in the main paper, and added more discussion to make this point clearer.
>
> 4. Dataset availability & more samples
>
> Since the dataset was collected from an industrial setting, we will not be able to release the full dataset as it is. However, in Appendix C, we have revised the paper to add more discussion of the dataset statistics. Specifically, we presented the distribution of formula lengths, and discussed the most commonly appearing operators.
>
> 5. Related work
>
> Thanks for pointing out these papers! These are very interesting works on neural program synthesis, and we have revised the related work section to include the discussion of these papers.

---

> > ### Comment · AnonReviewer1 · 2020-11-22
> > **Re: Clarification and discussion**
> >
> > Thanks for the response and the corresponding revision. Most of my questions have been addressed/discussed.
> >
> > **Formula accuracy**: I do understand the difficulty of enumerating those formulas that are semantically equivalent to the ground truth formula. Yet, the formula accuracy metric is not only just an underestimation but could also be "wrong". Specifically, the model performs worse based on the formula accuracy metric could actually produce more semantically correct formulas. I believe it would be important to make it clear in the paper.
> >
> > From my perspective, one possible way to address this would be (1) sampling a bunch of data points, (2) produce predicted output with the synthesized formula, (3) produce ground truth output with the ground truth formula, and (4) compare the predicted output to the ground truth output and compute the accuracy.  This sampling-based evaluation metric has been shown in "Neural Program Synthesis from Diverse Demonstration Videos (ICML 2018)", named execution accuracy, together with the formula accuracy (named sequence accuracy in the paper), providing both the upper bound and the lower bound of the performance. Yet, the difficulty of applying the sampling-based accuracy would be sampling a bunch of data points (i.e. a distribution of spreadsheet that requires the same formula). If possible, I encourage the authors to elaborate on this point and incorporate this into the paper to further improve it.
> >
> > I have carefully read the reviews from other reviewers and decided to keep my original rating (i.e. "7: Good paper, accept").

---

> > > ### Author Response · Authors · 2020-11-23
> > > **Follow-up discussion of the formula accuracy**
> > >
> > > Thanks for appreciating our work, and your thoughts on the evaluation metric! We agree that it is difficult to precisely evaluate the semantic equivalence in our case. Based on your suggestion, we revised Section 4.2 to add more related discussion, including the difficulty of evaluating the execution accuracy.

---

### Official Review · AnonReviewer2 · 2020-10-28
**Novel problem formulation**

**Rating:** 7
**Confidence:** 3

**Review:**

**Summary:**

This paper presents an interesting formulation for spreadsheet formula synthesis. Instead of taking the input output pairs as input, as is done in the programming by example (PBE) approaches, the proposed approach takes the semi-structured tabular context as input for predicting a formula for the target cell. A neural network architecture is presented which uses a BERT-based encoder to leverage the natural language meta-data.

**Strengths:**

* The paper is clear and well written. Problem formulation, model architecture and evaluation setup are presented in sufficient details.
* I found the program formulation quite interesting. For synthesizing a spreadsheet formula, instead of taking the input output pairs as input, the paper proposes to take the tabular context as input (without an explicit specification) . This is much closer to the real world application where spreadsheets are typically semi-structured and contain rich metadata.
* Supports the full  “Google Sheets Language” with rectangular ranges. This is much more complex than Flashfill and brings many challenges which the paper does well to address.
* The proposed approach out-performs the neural network approaches for programming by examples. It achieves reasonable accuracy on a large-scale benchmark to be usable in practice.
* Evaluation and ablation studies are quite thorough. In-depth analysis of results gives insight into the contribution of various architectural choices towards the overall result. The results are inline with the design rationale.
* The work is well situated and relation to the related work is discussed in sufficient details.

**Weaknesses:**

* I find the problem formulation interesting but would rate the overall approach as medium on novelty. The problem formulation is very specific to the task at hand and has a limited scope of being applicable to outside it (To be fair, the paper doesn’t claim so.) Nothing to take away, however, from the flawless execution.
* I couldn’t find any glaring weakness but here are a few questions/suggestions.
* Although the formulation takes into account the rich metadata of the semi-structured spreadsheets, it misses out on certain inputs that would have been easier for the user to provide. For example, users can provide the ranges as input. I believe this would improve the accuracy further since the sketch prediction accuracy is already higher. Have the authors tried something similar?
* It would help to add a few qualitative examples of synthesized formulas in the main paper to get the feel of the results.
* It is not clear to me if the output of the model/beam search always results in a valid program. It would also like to know the wall-clock time required for synthesis of top-5 formulas.
* Does the model come up with formulas that are semantically equivalent but syntactically different from the ground truth? How do you handle this in the evaluation?
* It is not clear to me if you handle formulas involving absolute as well as relative ranges. Are the absolute ranges part of the sketch in such cases?

**Overall Remark**

I believe that overall problem formulation and the architecture would be valuable to the community considering that the proposed system supports the full “Google Sheet Language” and performs reasonably well to be used in practice. The spreadsheet formula synthesis is an important domain with a wide audience.

---

> ### Author Response · Authors · 2020-11-21
> **Clarification and discussion**
>
> Thanks for appreciating our work and your insightful comments! We have incorporated your comments in our revision, and we respond to your questions and comments below.
>
> 1. Novelty of our approach
>
> We agree that our approach is designed for tasks with similar formulations to the spreadsheet formula prediction application evaluated in our work. However, there are various programming-by-spreadsheet applications where complex actions are described as formulas in spreadsheet tables, including CAD software and home-automation systems. For example, ShapeSheet spreadsheets are used to store the information of objects in Microsoft Visio, and these spreadsheets include formulas to determine the object attributes. Similarly, SketchUp allows users to import table data from a spreadsheet for 3D modeling. In our experiments, we evaluated on spreadsheets for general-purpose data processing, but our approach could also potentially be used for more specialized applications as mentioned above, where domain expertise is required and thus assistance is beneficial. In addition, we mentioned in Appendix A that our encoder design could potentially be adapted for other spreadsheet applications besides formula prediction, e.g., bug detection and clone detection.
>
> 2. Including the user-provided ranges as part of the model input
>
> Thanks for your great suggestion! We have indeed tried a similar approach. Specifically, for the decoding process, we feed in the ground truth cell ranges when the decoder is supposed to predict them. In this case, the accuracy becomes similar to the sketch accuracy. In fact, this is part of the motivation for presenting the breakdown accuracies on sketch and range prediction, because even partially correct formula predictions could be helpful in practice.
>
> 3. Providing more examples of model predictions in the main paper
>
> Thanks for your suggestion! We have revised the paper accordingly, and moved the sample predictions in the appendix to Section 4.3 in the main paper.
>
> 4. Beam search results and inference time
>
> For the beam search, we didn’t explicitly impose any constraints on the decoded formulas. Therefore, it is possible that the synthesized formula may not be valid, e.g., an operator may call cell values with data types not supported in the function definition. However, empirically we observe that with a sufficiently large dataset for training, the synthesized formulas by well-trained models are mostly valid. This is the case with both our full formula specification and the FlashFill-style specification. About the inference time, with TPU it takes 100ms for the beam size of 5.
>
> 5. Evaluation metric
>
> To measure the formula accuracy, because we only consider a predicted formula to be correct when it is exactly the same as the ground truth, for those predicted formulas that are semantically equivalent but different from the ground truth, they will be considered wrong in our evaluation. Indeed, sometimes the model predicts semantically equivalent but different formulas from the ground truth. For example, to predict arguments for SUM and MULTIPLY, the model could predict the cell ranges in a different order. However, it is hard to systematically define the semantic equivalence in our evaluation, because we aim to support a wide range of operators in the spreadsheet language. Therefore, we still focus on our current metric to measure the formula accuracy, even if it could be an under-estimate of the semantic equivalence. A similar discussion could be found in our response to R1 about the formula accuracy, and we have revised Section 4.2 in the paper to make this point clearer.
>
> 6. How to handle absolute ranges
>
> Our model only predicts relative cell references within a local range, and currently it does not support the prediction of absolute ranges. Augmenting our vocabulary with tokens for absolute cell references is a good choice, but we need to add a large number of tokens to systematically represent different possible cell positions. Thus, we focus on predicting relative cell ranges in this work, and we consider predicting other types of cell references as future work. We have revised Section 2 (the paragraph “The spreadsheet language”) and Appendix C to make this point clearer.

---

> > ### Comment · AnonReviewer2 · 2020-11-24
> > **Re: Clarification and discussion**
> >
> > Thank you for clarifying the questions and incorporating the suggestions.
> >
> > Nice to know that using the user-provided ranges improves the accuracy.  Also the inference time is low enough for the approach to be used in real applications. The accuracy is good enough and will only go further up if more sophisticated definition of semantic equivalence is incorporated.
> >
> > Overall, I believe that the paper will be valuable contribution to the community. I would like to retain my original rating (7: Good paper, accept).

---

### Official Review · AnonReviewer3 · 2020-10-28
**Summary: This paper presents a technique to generate formulas given table context in spreadsheet (even though I am not very clear about motivation and usecase). Authors propose a BERT based model to learn dependencies  between table rows and columns.**

**Rating:** 3
**Confidence:** 3

**Review:**

The motivation for the work is not very clear to me. The paper is difficult to follow especially the motivation, problem setup and architecture part. The dataset is not publicly available for everyone to use to the idea is not reproducible. Authors write in the paper that dataset is publicly available in their organization, I don’t understand what do they mean by that. Comparison with state of the art neural symbolic machine/ function generators is missing. I didn’t find the baselines to be strong enough.

Overall:  I found this paper to be hard to follow. The motivation for the task of predicting formulas given table is not very clear. It would have been better if authors could explain a real world example application for their task. Figure-1 is not clear, it needs some more intuition and discussion. A running example would have really helped. I found introduction to be a bit disconnected.

Question: why authors have not reported results for other related tasks like taskfill using their technique? Can you please explain motivation/real world application for the task and technique in details? How different their taks is from semantic parsing for table question answering?

---

> ### Author Response · Authors · 2020-11-13
> **Please provide more concrete suggestions on how to improve our work**
>
> Thanks for your review. We are happy to address any concerns regarding our work. However, it would be helpful if you could explain your comments with some more details.
>
> 1. “The motivation for the task of predicting formulas given table is not very clear. It would have been better if authors could explain a real world example application for their task.”, “Can you please explain motivation/real world application for the task and technique in details?”
>
> We have explained the real-world applications in Section 1 (Introduction) and Section 2 (Problem Setup). Our approach and problem setup could be used for predicting formulas in spreadsheets, e.g., those written in Google Sheets. We want to do this because many end-users of spreadsheets may not be familiar with spreadsheet formulas. We want to automatically suggest formulas to a user that they may want to use, based on the data that they have already entered into the spreadsheet. Even for expert users, writing the desired formulas can sometimes take a long time as they may not know the exact Formula syntax to use. We demonstrated the results of our system that automatically predicts formulas on a dataset of real-world spreadsheets.
>
> 2. “The paper is difficult to follow especially the motivation, problem setup and architecture part.”
>
> Could you please point us to the specific parts that were difficult to follow? We are happy to revise our paper and provide more details if anything is missing.
>
> 3. “The dataset is not publicly available for everyone to use to the idea is not reproducible. Authors write in the paper that dataset is publicly available in their organization, I don’t understand what do they mean by that.”
>
> Google Sheets enables users to share their documents with other users within the same organization, e.g., the same university or the same company. Our dataset includes spreadsheets that are shared within our organization. Since the dataset was collected from an industrial setting, we will not be able to release the full dataset as it is. However, our modeling technique could also be applied to datasets from other organizations.
>
> 4. “Comparison with state of the art neural symbolic machine/ function generators is missing. I didn’t find the baselines to be strong enough.”
>
> We have compared with strong baselines for related tasks, including: (1) RobustFill, the state-of-the-art neural network approach for the FlashFill task; and (2) different variants of models with BERT, where BERT-based models have achieved the state-of-the-art for many semantic parsing and question answering benchmarks.
>
> 5. “Figure-1 is not clear, it needs some more intuition and discussion. A running example would have really helped.”
>
> In Section 2, we provide a detailed description of how to decode the formulas in Figure 1. Please let us know what specific questions you would like us to discuss more in the paper and/or in the response.
>
> 6. “why authors have not reported results for other related tasks like taskfill using their technique?”
>
> To our knowledge, we don’t know if there is any task called “taskfill”. Feel free to include the reference for “taskfill”. If you mean “FlashFill” instead, we presented the results for comparison with RobustFill in Section 4.3.3.
>
> 7. “How different their task is from semantic parsing for table question answering?”
>
> We have discussed the differences in Section 5 on page 8, with a whole paragraph starting from “In terms of the model input format, our spreadsheet formula prediction task is related to existing benchmarks on semantic parsing over a tabular database”. As noted in that paragraph, there are two main differences: (1) table question answering includes a question in the program specification, while our task does not, and thus our program specification is more ambiguous; and (2) our spreadsheet tables are typically less structured than the tables for question answering tasks.
>
> We request you to kindly reconsider your review in light of this discussion. If you have more specific questions, we are happy to provide more clarification and revise our paper accordingly.

---

### Author Response · Authors · 2020-11-21
**Response and revision**

We thank reviewers for their insightful feedback and appreciating our work! We have revised the paper with the following major changes to incorporate the comments.

1. Based on R2’s suggestion on adding more qualitative examples of synthesized formulas in the main paper, we moved sample predictions from the appendix to Section 4.3 in the main paper.  Based on R1’s comments, we also added discussion of these examples in the main paper, to emphasize that our formula specifications, e.g., the headers, are inherently noisy.

2. Based on R1 and R2’s questions about the evaluation metric, we have revised Section 4.2 and added more discussion of our formula accuracy metric.

3. Based on R1’s question on dataset statistics, in Appendix C, we added the description of formula length distribution, and discussed the commonly used spreadsheet functions and operators.

4. Based on R2’s question, we have revised Section 2 (the paragraph “The spreadsheet language”) and Appendix C to make it clearer that we do not handle absolute ranges in this work.

5. Based on R1’s suggestion on the related work discussion, we have revised the related work section, and added the discussion of the suggested papers on neural program synthesis for different tasks.

---

### Decision · Program_Chairs · 2021-01-07
**Final Decision**

**Decision:**

Reject

**Comment:**

This paper clearly has great ideas and reviewers appreciated that. However, the lack of experiments that can be validated by the community (only 1 experiment on the proprietary dataset) is an issue. We don't know if the reported accuracy is a respectable one (in the public domain).   Having a proprietary dataset is a plus, but no public benchmark raises concerns about reproducibility.
We recommend the authors to add some tasks and benchmarks for the community to check for themselves that the numbers reported are non-trivial.